# SWATH-MS Based Proteomic Profiling of Prostate Cancer Cells Reveals Adaptive Molecular Mechanisms in Response to Anti-Androgen Therapy

**DOI:** 10.3390/cancers13040715

**Published:** 2021-02-09

**Authors:** Chamikara Liyanage, Adil Malik, Pevindu Abeysinghe, Judith Clements, Jyotsna Batra

**Affiliations:** 1Faculty of Health, Institute of Health and Biomedical Innovation, School of Biomedical Sciences, Queensland University of Technology, Brisbane, QLD 4059, Australia; chamikara.liyanage@hdr.qut.edu.au (C.L.); m.malik@hdr.qut.edu.au (A.M.); abeysinghe.abeysingh@hdr.qut.edu.au (P.A.); j.clements@qut.edu.au (J.C.); 2Australian Prostate Cancer Research Centre-Queensland (APCRC-Q), Translational Research Institute, Queensland University of Technology, Brisbane, QLD 4012, Australia

**Keywords:** prostate cancer, proteomics, transcriptomics, anti-androgen therapy, bicalutamide, enzalutamide

## Abstract

**Simple Summary:**

Androgen targeted therapy has been foundational in the management of advanced prostate cancer. Nevertheless, responses to the therapy are found to be seldom sustained, in which patients develop resistance and progress to a lethal and incurable castration-resistance stage. Therefore, comprehensive understanding of the molecular basis of treatment induced cellular responses is required to circumvent molecular mechanisms driving castration-resistance. Using an advanced, robust quantitative proteomic analysis, we profiled the prostate cancer cell proteome induced by the two AR antagonists/anti-androgens, Bicalutamide and Enzalutamide. We highlighted key molecular signatures and cellular pathways that provide insights into the anti-androgen induced adaptive cellular programming in prostate cancer cells. Targeting these molecules and associated pathways might be useful in developing novel therapeutic approaches and/or as biomarkers of predicting prostate cancer treatment response.

**Abstract:**

Prostate cancer (PCa) is the second most common cancer affecting men worldwide. PCa shows a broad-spectrum heterogeneity in its biological and clinical behavior. Although androgen targeted therapy (ATT) has been the mainstay therapy for advanced PCa, it inevitably leads to treatment resistance and progression to castration resistant PCa (CRPC). Thus, greater understanding of the molecular basis of treatment resistance and CRPC progression is needed to improve treatments for this lethal phenotype. The current study interrogated both proteomics and transcriptomic alterations stimulated in AR antagonist/anti-androgen (Bicalutamide and Enzalutamide) treated androgen-dependent cell model (LNCaP) in comparison with androgen-independent/castration-resistant cell model (C4-2B). The analysis highlighted the activation of MYC and PSF/SFPQ oncogenic upstream regulators in response to the anti-androgen treatment. Moreover, the study revealed anti-androgen induced genes/proteins related to transcription/translation regulation, energy metabolism, cell communication and signaling cascades promoting tumor growth and proliferation. In addition, these molecules were found dysregulated in PCa clinical proteomic and transcriptomic datasets, suggesting their potential involvement in PCa progression. In conclusion, our study provides key molecular signatures and associated pathways that might contribute to CRPC progression despite treatment with anti-androgens. Such molecular signatures could be potential therapeutic targets to improve the efficacy of existing therapies and/or predictive/prognostic value in CRPC for treatment response.

## 1. Introduction

Despite the advances in prostate cancer (PCa) diagnosis and treatment, temporal trends of PCa incidence and mortality varied significantly globally during the past few years [1]. Although, localized disease is easily managed with curative surgery or radiotherapy, some patients recur systemically with advanced metastatic PCa. Due to the androgen-dependency and the crucial role of the androgen receptor (AR) in PCa progression, androgen deprivation therapy (ADT) has been denoted as the mainstay therapy for relapsed or metastatic patients [2]. ADT has been progressively supplemented with inhibitors of androgen synthesis and AR antagonists/anti-androgens, therefore termed as androgen targeted therapy (ATT) [3]. Although ADT/ATT is initially effective, many patients invariably fail to respond after ~13 months and eventually progress to a lethal stage: androgen-independent/castration-resistant PCa (CRPC) [4].

PCa is an extremely heterogeneous disease, hence understanding the cellular basis of cell heterogeneity is important to comprehend the ADT/ATT resistance during CRPC progression. Complex AR-dependent adaptive mechanisms have been identified that cause reactivation of AR signaling enabling the continued growth and progression of CRPC. This includes AR variants with increased sensitivity to residual androgens after castration and overexpression of AR or downstream effectors via ligand-independent modifications [5]. Despite the current findings of AR signaling in CRPC, AR-independent molecular mechanisms involved in treatment resistance and the paradigm shift of hormone-naive PCa leading to CRPC remain largely unknown. Therefore, a broader understanding of the cellular functioning during ADT/ATT is essential as it might lead to novel or co-treatment strategies to improve efficacy of ATT. Although previous studies have investigated the changes in androgen-responsive genes (ARGs) towards PCa progression, only a few attempts have been taken to explain off-target effects of AR antagonists/anti-androgens that bypass AR signaling and other AR-independent mechanisms [6,7].

In the current study, we aimed to delineate AR antagonists/anti-androgens induced molecular alternations in PCa cells. We used androgen-dependent (AD) cell model (LNCaP) treated with 1st generation non-steroidal anti-androgen: Bicalutamide (BIC) and 2nd generation non-steroidal anti-androgen: Enzalutamide (ENZ). In order to profile the PCa proteomic changes, we have used the sequential window acquisition of all theoretical mass spectra (SWATH-MS) label-free quantification method, and RNA sequencing (RNAseq) analysis was used to observe the changes in PCa transcriptome. Comparative analysis between LNCaP cells and the metastatic, androgen-independent (AI) cell model (C4-2B) represented molecular changes acquired during the paradigm shift of androgen-dependent PCa to androgen-independence. Ultimately, the study highlighted key molecular alternations and pathways that emerged during anti-androgen therapy, supporting treatment escape and PCa progression to CRPC.

## 2. Results

### 2.1. Summary of the Proteomic and Transcriptomic Profiling of PCa Cell Lines

To mimic the anti-androgen therapy given for patients in the androgen-dependent stage and find alternations in the PCa cell proteome, LNCaP cell line was obtained and supplemented with androgen/dihydrotestosterone (DHT), BIC and ENZ treatments (Appendix A, see Section 4). Via SWATH-MS proteomic profiling of treated LNCaP cells, a total of 2628 proteins were identified at 1% False discover rate (FDR). According to the differential expression analysis we identified 318, 215 and 182 significantly differentially expressed proteins (DEPs) for DHT, DHT + BIC and DHT + ENZ treatments, respectively (Figure 1A). A total of 74 DEPs were found to be shared between all three treatments, while 75 and 51 DEPs were found specific to DHT + BIC and DHT + ENZ treatment, respectively. In addition, 319 and 258 DEPs were detected with BIC and ENZ only treatments, providing treatment specific protein profiles stimulated under direct inhibition of AR signaling (Figure 1B). To mimic the PCa progression from androgen-dependent to castration-resistant stage, we obtained the C4-2B cell line: a metastatic derivative of the LNCaP cell model. SWATH-MS proteomic profiling of the C4-2B cells has identified 2703 proteins at 1% FDR. Differential expression analysis has reported 159 DEPs compared to LNCaP cells cultured in growth factor supplement conditions (LNCaP-FBS) and 208 DEPs compared to LNCaP cells cultured under hormone/androgen depleted conditions (LNCaP-CSS) (Figure 1C). Finally, the common feature of DEPs was the predominance of up-regulated protein expression in response to the treatments, particularly in BIC (69.4%) and ENZ (75.0%) only treatments (Figure 1D). Moreover, a majority of the proteins were found up-regulated in C4-2B cells compared to LNCaP-FBS or LNCaP-CSS cells. (Figure 1D).

Both principal component analysis (PCA) and hierarchical clustering analysis demonstrated the distinct clustering between DHT and DHT + BIC/ENZ treatment proteomic profiles (Figure 2A,B). DHT + BIC/ENZ treatments were clustered separately from the BIC/ENZ only treatments, suggesting the differential effect of anti-androgens on PCa cell behaviour, depending on androgen availability (Figure 2A,B). Moreover, a clear separation between LNCaP and C4-2B proteomic profiles was observed, signifying distinct molecular programs characterized between AD and AI PCa phenotypes (Figure 2C,D).

RNAseq analysis has identified a total of 19,549 genes in LNCaP cells. A total of 2556 common proteomic and transcriptomic elements were reported allowing a better integration (~97%) between the two datasets (Figure 3A). Although, linear regression analysis showed a positive correlation between the two datasets, the high variability (low R^2^) indicated the limited concordance between transcriptomic and proteomic expressions in PCa cells (Figure 3B). Differential expression analysis has identified a total of 2377, 433 and 503 significantly differentially expressed genes (DEGs) in DHT, DHT + BIC and DHT + ENZ treatments, respectively (Appendix A). A total of 81 DEGs were found to be shared between all three treatments, whereas 137 and 201 DEGs were specific to DHT + BIC and DHT + ENZ treatments, respectively. In contrast to the proteomic data, transcriptomic data showed that majority of the genes found downregulated (70.7%) with the DHT treatment (Appendix A). In line with the observations of proteomic profiling, hierarchical clustering has given a clear separation between DHT and DHT + BIC/ENZ transcriptomic profiles (Appendix A).

### 2.2. Effects on AR Signaling during Anti-Androgen Treatment

Since growth and proliferation of PCa cells largely depends on AR signaling, we characterized the AR signaling activities of LNCaP cells in response to anti-androgen treatments. In line with previous studies, our study reported AR expression was unaltered in response to both anti-androgens, at mRNA and protein levels (Figure 4A,B) [8,9]. However, AR mediated transcriptional activation of ARGs (MSigDB Hallmark “Androgen Response” gene set) were found to be suppressed in DHT + BIC/ENZ treatments compared to the DHT treatment (Figure 4C). A similar observation was made at the protein level in which protein expression of four ARGs: KLK3, FKBP5, NDRG1, and UAP1 were found to be supressed with DHT + BIC/ENZ and BIC/ENZ only treatments compared to DHT treatment (Figure 4D). The greater reduction of the ARG expression with ENZ treatment (with or without DHT stimulation) may indicate stronger AR-antagonist activity of ENZ compared to BIC treatment. Although not statistically significant, we observed downregulation of proteins related homologous recombination (HR): ATM, RAD51, FANCA, CHEK2 and upregulated PARP1 protein expression in response to BIC and ENZ, demonstrating suppressed HR activity and activated PARP DNA repair system during anti-androgen treatment (Appendix A).

### 2.3. Androgen and Anti-Androgen Mediated Molecular Networks and Biological Processes in PCa Cells

Androgen and anti-androgen mediated disrupted gene/protein networks were identified by the IPA analysis, hinting at adaptive molecular pathways supporting PCa cell survival and proliferation. We focused on the top scoring cancer related protein/gene networks induced by androgen/anti-androgen treatments showing a functional relationship between DEPs/DEGs (Appendix A). Under DHT treatment, the analysis showed a significant enrichment in the protein network of RNA post-transcriptional modification (Score = 67, DEPs = 30) and the gene network of cellular movement (Score = 34, DEGs = 32) (Appendix A). In addition, DHT + BIC treatment showed an overrepresentation in the protein network of RNA-post transcriptional modifications (Score = 60, DEPs = 30) and the gene network of molecular transport (Score = 27, DEGs = 18) (Appendix A). Moreover, DHT + ENZ treatment demonstrated a significant enrichment in the protein network of protein synthesis and RNA damage and repair (Score = 63, DEPs = 30) and the gene network of post-translational modification (Score = 34, DEGs = 22) (Appendix A).

To identify biological processes significantly dysregulated (FDR/q < 0.05) during androgen-anti-androgen treatment, we overlapped DEGs/DEPs into Gene Ontology Biological Processes (GO-BP), using Gene Set Enrichment Analysis (GSEA) in Molecular Signatures Database (MSigDB) (Appendix A) [10]. A total of GSEA analysis revealed that, up-regulated molecules in DHT treatment were mainly associated with regulation of mRNA metabolism (DEPs, FDR = 5.06 × 10^−72^, Figure 5A) and small molecule metabolism (DEGs, FDR = 1.72 × 10^−47^
Appendix A), whereas down-regulated molecules in DHT treatment showed enrichment in intracellular transport (DEPs, FDR = 2.35 × 10^−53^, Figure 5B) and biological adhesion (DEGs, FDR = 1.24 × 10^−46^, Appendix A). Up-regulated molecules in DHT + BIC treatment were found to affect RNA processing (DEPs, FDR = 1.39 × 10^−74^, Figure 5C) and protein coupled receptor signaling (DEGs, FDR = 3.79 × 10^−9^, Appendix A) whereas intracellular transport (DEPs, FDR = 9.28 × 10^−191^, Figure 5D) and ion transport (DEGs, FDR = 5.75 × 10^−5^, Appendix A) were affected by down-regulated molecules in DHT + BIC treatment. Moreover, the majority of up-regulated molecules in DHT + ENZ treatment were related to RNA processing (DEPs, FDR = 9.8 × 10^−59^, Figure 5E) and ion homeostasis (DEGs, FDR = 4.66 × 10^−12^, Appendix A), while down-regulated molecules in DHT + ENZ treatment were involved in intracellular transport (DEPs, FDR = 1.43 × 10^−28^, Figure 5F) and biological adhesion (DEGs, FDR = 5.01 × 10^−9^, Appendix A). Overall, the analysis highlighted the key effect of anti-androgen on RNA metabolism, indicating the distinct transcriptional programming in PCa cells activated under the anti-androgen induced stress conditions.

### 2.4. Upstream Regulators Dysregulated in Response to the Anti-Androgen Treatment

Next, we performed upstream regulatory analysis to identify likely affected transcription regulators during androgen and anti-androgen treatments (Appendix A). Intriguingly, the MYC upstream regulator was found activated in both DHT + BIC (activation Z-score: 3.188, *p* = 1.14 × 10^−09^) and DHT + ENZ (activation Z-score: 2.849, *p* = 8.11 × 10^−06^) treatments, while it was found inhibited in response to DHT treatment (activation Z-score: −2.678, *p* = 1.35 × 10^−16^) (Appendix A, Figure 6A,B). Both DHT + BIC/ENZ treatments have induced the upregulation of several MYC targets, mainly ribonucleoproteins (HNRNPs, SNRNP), pre-mRNA splicing factors (PRPs) and DEAD-BOX (DDX) RNA helicases that regulate RNA metabolism. Moreover, MYC oncogenic targets such as DEK and YBX1 proteins were found upregulated in response to both DHT + BIC/ENZ treatments. In addition to MY C, activation of proline and glutamine-rich (PSF/SFPQ) RNA binding/splicing factor upstream regulator was observed in both DHT + BIC (activation Z-score: 2.429, *p* = 3.65 × 10^−08^) and DHT + ENZ (activation Z-score: 2.216, *p* = 9.5 × 10^−06^) while no change in PSF/SFPQ activation/inhibition was observed in DHT treatment (Appendix A, Figure 6C,D). Expression of several PSF/SFPQ target splicing factors, such as HNRNPU, SF3B2 and U2AF1/2 were found upregulated in response to both BIC and ENZ treatments. Overall, this analysis shed light on the differential expression of numerous mRNA-splicing factors regulated by MYC and PSF/SFPQ upstream regulators during anti-androgen treatment.

### 2.5. Anti-Androgen Induced Molecular Pathways That Denote the Metastatic Androgen Independent PCa Phenotype

Next, we sought to determine enrichment of treatment induced DEPs in molecular pathways that might provide PCa cells with a growth advantage. DEPs induced by DHT + BIC/ENZ treatments and DEPs identified in C4-2B vs. LNCaP comparison were mapped in to KEGG pathways. Interestingly, the analysis revealed that pathways dysregulated in the androgen-independent C4-2B cell model were comparable with those dysregulated in DHT + BIC/ENZ treatments (Appendix A).

We observed a significant enrichment of DEPs in genetic information processing pathways, such as (i) spliceosome pathway: HNRNPU/K; YBX1; SRSF2/3/7; DDX5, SNRNP70 (Appendix A), (ii) ribosome biogenesis: DKC1 (Appendix A), (iii) ribosome: RPL17/19/29; RPS10/17/19 (Appendix A), (iv) RNA transport pathway: SRRM1; translation initiation factors (eIFs) (Appendix A) and (v) Protein processing factors in ER: SSR3; DAD1 (Appendix A). Moreover, proteins related to energy metabolism: oxidative phosphorylation (OXPHOS) were found dysregulated, mainly the proteins involved in NADH dehydrogenase activity: NDUFA2/10; NDUFS2 (Appendix A). In addition, proteins involved in cell growth, death and cell communication pathways were found dysregulated, such as (i) MCM proteins related to cell cycle maintenance and (ii) PAK4 associated with cellular focal adhesion (Appendix A). Furthermore, proteins that belong to the PI3K-AKT and MAPK signaling pathway, such as MAPK1 were found dysregulated (Appendix A). A predominant involvement of cancer related molecular alterations was observed, for instance, (i) overrepresentation of AR transcriptional activity reported by the upregulation of AR targets: KLK3 and heat shock protein (HSPs): HSP90AB1 (Appendix A) and (ii) enrichment of cancer associated proteoglycans, such as Cortactin (CTTN) (Appendix A). Overall, the current analysis highlighted a set of proteins related to 11 cancer related signaling pathways that might aggravate treatment resistance and lead to the insurgence of androgen-independence (Appendix A, Figure 7).

### 2.6. Validation of Gene/Protein Expression Using PCa Clinical Data

We have utilized publicly available PCa transcriptomic and immunohistochemical (IHC) datasets to verify the expression of the two upstream regulators and key oncogenic genes/proteins in clinical samples. Based on PCa patient transcriptomic data retrieved from Oncomine^TM^ database, we observed the marked upregulation of the *SFPQ* upstream regulator in CRPC tissues compared to localized PCa and normal prostate tissues. Although, the *MYC* upstream regulator was upregulated in localized PCa compared to normal prostate tissues, no significant change was observed in CRPC tissues. In addition, *HNRNPU*; *DKC1*; *CTTN*; *MCM2*; *MAPK1*; *PAK4* genes were found to be significantly upregulated in CRPC tissues compared to normal prostate tissues (Figure 8). Based on the IHC data retrieved from the HPA database, both MYC and SFPQ upstream regulators showed a moderately higher protein expression in high-grade PCa tissues compared to the low-grade PCa tissues. Likewise, HNRNPU, MCM2, MAPK1, CTTN protein targets also demonstrated relatively higher expression in high-grade PCa tissues compared to the low-grade PCa tissues (Figure 9). Intriguingly, DKC1 showed a high expression in both high-/low-grade tissues, whereas PAK4 was expressed at moderate levels in both high-/low-grade PCa tissues. In summary, the above analysis highlighted on the possible involvement of anti-androgen mediated upstream regulators and oncogenic proteins in PCa pathogenesis and their effect on fueling PCa progression to CRPC.

## 3. Discussion

Identifying molecular alterations dysregulated during ATT which likely leads to treatment resistance could offer new opportunities for patient selection and management, preventing disease progression to CRPC. The current study investigated both transcriptomic and proteomic changes induced by BIC and ENZ anti-androgen treatments that might be responsible in developing PCa treatment resistance and androgen-independence.

We increased the depth of proteomic profiling in our cell models via SWATH-MS strategy: an accurate, highly reproducible label-free approach in whole proteome quantification [11,12]. Two recent studies that used SWATH-MS strategy on LNCaP cells validated the total number of proteins identified in our study [13,14,15]. In addition, we observed a limited correlation between PCa transcriptomic and proteomic datasets, in agreement with previous PCa omics-based research [16,17]. Varied post-transcriptional modifications (PTMs) associated with translation regulation and kinetic differences between protein synthesis and turnover may explain the poor correlation between mRNA and protein abundances in complex biological samples [18].

Although, CRPC is frequently associated with the loss of AR expression, recent studies describe that anti-androgen resistance and PCa progression can be acquired via an AR indifferent state, where cancer cells do not depend on AR signaling regardless of persistent AR expression [8,9]. We observed that anti-androgen treatment blocks AR mediated transcriptional activities without affecting AR expression, which provides evidence for the potential of LNCaP cells to acquire AR-independent adaptive mechanisms via an AR-indifferent state. The strong AR-antagonistic activity shown by ENZ can be explained by its higher affinity for AR in comparison to BIC [19]. The clinical benefit of ENZ over BIC in patients with metastatic CRPC and its superior activity in suppressing AR signaling has been demonstrated in clinical trials [20,21]. Nevertheless, AR signaling can be continued in a fraction of cells due to the potential agonistic functioning of BIC and ENZ, which allow LNCaP cells to maintain their dependency on AR signaling [8,22,23]. This observation is clinically manifested by the anti-androgen withdrawal syndrome and clinical improvement observed upon anti-androgen discontinuation [22]. Moreover, our data demonstrated moderate downregulation of HR gene expression and moderate upregulation of PARP activity in response to both BIC and ENZ treatments. Although expression levels are not statistically significant direction of HR and PARP protein regulation is in line with previous observations that show PARP backup DNA repair system is activated in absence of HR which support the survival of PCa cells during ADT [23,24].

Cancer cell survival depends on the coordinated activation of translation initiation factors and major signaling cascades regulating translation that are implicated in a broad range of cellular processes [25,26]. Our analysis highlighted the overrepresentation of molecular networks associated with transcriptional and translational regulation, which might allow them to adapt to anti-androgen induced cellular stress. This finding was further confirmed by the MYC and PSF/SFPQ mediated transcriptional activation of splicing factors that regulate the expression of genes promoting PCa survival. A previous study confirmed that androgen-treatment in hormone-depleted PCa cells reduced MYC levels [27]. MYC plays an important role in regulating AR and its variant expression through transcriptional regulation coupled with modulation of protein stability [28]. Furthermore, it has been demonstrated that BIC has the potential to reverse androgen-mediated reduction of MYC [29]. Therefore, the transcriptional regulation conferred by MYC activation is considered to be a mechanistic insight into the ability of PCa tumors to overcome stressful conditions induced by anti-androgen therapy [30,31,32]. On the other hand, PSF/SFPQ mRNA splicing factor has been previously reported to be involved with the worse prognosis of PCa as shown by the upregulation of PSF/SFPQ downstream targets in metastatic PCa patients [33]. These splicing factors are responsible for AR overexpression and alternative-splicing (AR-v7) which are key drivers towards androgen-independence [33]. Hence, upstream MYC and PSF/SFPQ regulators and downstream transcription factors have an obvious pharmacological importance as their inhibition could be a viable approach to alleviate anti-androgen resistance [28].

We observed the coordinated activation of pathways in LNCaP cells treated with anti-androgens and their overlap with those dysregulated in C4-2B cell populations. Key oncogenic proteins dysregulated in these pathways represent the molecular repertoire that support treatment escape and progression to CRPC. The highly perturbed environment shaped by the androgen-depleted medium supplemented with anti-androgens has triggered molecular pathways related to gene transcription and translation regulation. This includes the dysregulation of spliceosome components, such as HNRNPs, YBXs, SRSFs, DDXs and SNRNPs, that act as proto-oncogenic splicing regulators involved in the adaptive transcriptional programming of hormone-related cancers, such as breast and PCa [34,35,36,37,38]. HNRNPU and HNRNPK are two splicing factors involved in regulating AR and AR-v7 expression by post-transcriptional mechanisms and it has been previously found that BIC treatment alters AR/HNRNPK interaction and AR activation [35,39,40]. Proteins involved in ribosome biogenesis: DKC1 and ribosomal components: RPL19/29, RPS19/20 have been previously reported for their dysregulation and malignant potential in advanced PCa phenotypes [41,42]. Splicing factor: SRRM1 that is involved in the RNA transport pathway has been recently described for its association with PCa proliferation/migration-rate via modulating AR-v7 levels [39]. ER signal sequence receptor subunit: SSR3 that regulates protein entry into ER plays a major role in protein processing in ER and its dysregulation has been found associated with tumor growth in PCa [40,43,44]. DAD1 protein is involved in protein glycosylation and translocation across the ER that exhibit anti-apoptotic functions [45]. Previous data shows the upregulated expression of DAD1 in AI-PCa cell lines and grade-associated expression in PCa clinical tissues, thus, providing functional insights of DAD1 in therapy resistance and PCa progression [46,47].

Our data showed the dysregulation of OXPHOS, particularly characterized by upregulated NADH dehydrogenase activity which is the rate-limiting step in overall respiration. Overrepresentation of OXPHOS provides PCa cells with a growth advantage by overproduction of ROS to cause more DNA damage and mutations [48,49]. This observation is in line with previous data that propose OXPHOS involvement in androgen-independence [50,51]. Overexpression of minichromosome maintenance proteins (MCM) involved in cell cycle regulation are currently used as proliferation markers to determine tumor growth propensities [52,53,54]. Dysregulation of MCM proteins observed in our findings suggest their role in facilitating tumor cell proliferation during anti-androgen treatment. Upregulation of proteins involved in focal adhesion, such as PAK1 indicate the coordinated activation of cancer cell invasion in response to anti-androgen treatment [55,56,57]. Reiterating previous literature, our data showed the dysregulation of proteins belonging to the PI3K-AKT and MAPK pathway, such as MAPK1. Activation of PI3K-AKT/MAPK signaling promotes PCa cell proliferation and survival, hence combining PI3K-AKT/MAPK inhibitors with anti-androgen therapy were found to be promising in clinical trials [58,59]. Upregulation of KLK3/PSA in C4-2B cells confirms the activation of AR-signaling during PCa progression to CRPC via various AR bypassing mechanisms [60,61,62]. Moreover, overexpression of PSA can induce a feedback regulatory loop by which AR expression is maintained during CRPC progression [63]. On the other hand, HSPs, which play a pivotal role in AR stabilization during AR signaling, were found upregulated in response to anti-androgen treatment and in CRPC cells. Previous literature suggests that BIC agonistic AR activation and its downstream interaction with coactivators and transcriptional factors may rely on the stabilization provided by HSP90s [64]. Hence, using HSP inhibitors (e.g., geldanamycin) in combination with anti-androgens would increase the efficiency of PCa therapy. Proteoglycans are key constituents of extracellular matrix capable of modulating cancer cell invasion [65]. Therefore, it signifies the contribution of upregulated proteoglycans, such as CTTN towards PCa cell survival and invasion during anti-androgen therapy.

The limited correlation between the proteomic and transcriptomic data might limit the identification of extensive functional changes in PCa cells in response to stress conditions. Thus, understanding the limitations between different measurement techniques is critical before interpreting the results integrating datasets from multiple sources [66]. SWATH-MS DIA data acquisition is the easiest compared to DDA acquisition methods, yet difficult in terms of data analysis as it requires large informatic resources and sophisticated software tools [67]. Moreover, protein identification in SWATH-MS strategy is largely limited by the spectral library composition; therefore, generating a larger-scale spectral library is essential using both cell line and tissue DDA data. Protein targets identified in the current study are highly likely to be biomarkers for CRPC as well as targets for novel therapies. Nevertheless, our findings require further validation by applying different AD/AI cell lines and molecule/pathway inhibitors, before testing them using in vivo mouse models. Based on the resistance observed for BIC and ENZ, novel anti-androgens (e.g., Apalutamide, Darolutamide) have already made their way through preclinical and clinical studies to treat both metastatic and non-metastatic CRPC [68]. Therefore, the next wave of scientific focus should be identifying possible adaptive mechanisms induced by these novel anti-androgens to alter the course of PCa treatment resistance.

## 4. Materials and Methods

### 4.1. Cell Culture and Androgen/Anti-Androgen Treatment

LNCaP and C4-2B cell lines were obtained from the American Type Culture Collection (ATCC). Both cell lines were authenticated by Short tandem repeat (STR) profiling and tested negative for *Mycoplasma*. LNCaP cells were seeded in 6-well plates using RPMI-1640 media (Life Technologies, Mulgrave, Australia) supplemented with 5% fetal bovine serum (FBS, Sigma-Aldrich, Sydney, Australia) and were incubated at 37 °C for 48 h. The medium was replaced with androgen-depleted culture medium (RPMI1640) containing 5% charcoal-striped serum (CSS) for 48 h. Next, cells were supplemented with androgens: DHT (10 nM DHT, Sigma-Aldrich), or anti-androgens: BIC and ENZ (10 μM, Selleckchem.com, Waterloo, Australia) and Ethanol (EtOH)/vehicle control (20%, HPLC grade, Sigma-Aldrich), and incubated at 37 °C for additional 48 has described previously [69]. C4-2B cells were cultured in RPMI-1640 media supplemented with 5% FBS.

### 4.2. Liquid Chromatography Tandem Mass Spectrometry (LC-MS/MS)

#### 4.2.1. Sample Preparation

Cell pellets were first washed with 1X phosphate-buffered saline (PBS) 3 times and lysed using Sodium deoxycholate (SDC) buffer (1% SDC in 1 M Tris pH 8.0). Cell lysates were centrifuged at 16,000 rpm for 20 min to remove the cell debris and other contaminants. The samples were then sonicated in an ultrasonic bath (Thermo Scientific™, Waltham, MA, USA) for 15 min (at 4 °C, 100% Power) to denature proteins and shear DNA. The soluble fraction was collected after a centrifugation step at 1000× *g* for 15 min. The concentration of proteins was calculated using Bicinchoninic acid assay (BCA) with Pierce™ Bovine Serum Albumin (BSA) Standards (Thermo Scientific™). Of the protein extract, 10 μg was denatured at 95 °C for 5 min using a thermomixer (Eppendorf ThermoMixer^®^ F1.5, Hamburg, Germany). Denatured protein samples were reduced with 10 mM Tris (2-carboxyethyl) phosphine (TCEP) (Sigma-Aldrich), for 30 min at room temperature, and alkylated in the dark with 40 mM 2-chloroacetamide (2CAA) (Sigma-Aldrich) for 30 min at room temperature. Samples were then digested overnight at 37 °C after adding trypsin (Sigma-Aldrich) at a 1:50 enzyme-protein ratio. The reaction was quenched by 10% Trifluoroacetic acid (TFA) (Sigma-Aldrich), and spun down at 14,000 RPM to precipitate and remove excess SDC. Next, peptides were desalted using Pierce™ C18 Spin Tips (Thermofisher, Waltham, MA, USA), washed in 0.1% TFA, and eluted in to 80% acetonitrile (ACN) (HPLC grade, Sigma-Aldrich). Solvents were evaporated in a SpeedVac centrifuge (Savant Speed Vac, SPD121P-230, Thermo Electron Corporation, Milford, MA, USA) at 35 °C and re-suspended using Buffer A, which includes 2% ACN, 0.1% TFA spiked with iRT calibration mix (Biognosys AG, Schlieren, Switzerland). Peptide mixtures were transferred in to HPLC vials (Thermofisher). Sample preparations were performed in 3 biological replicates.

#### 4.2.2. Data Dependent Acquisition (DDA)

LC-MS/MS analysis was performed by the Central Analytical Research Facility (CARF). Peptide solutions were analysed on an Agilent 1260 HPLC system (Agilent Technologies, Santa Clara, CA, USA) coupled to a TripleTOF 5600+ mass spectrometer with NanoSource III (ABSciex, Toronto, Ontario, Canada). For each sample, 1 μg of peptides was loaded onto a ZORBAX C18 trap column (Agilent Technologies, Santa Clara, CA, USA). Loaded materials were then eluted from an analytical column (75 μm × 15 cm) with an integrated manually pulled tip packed with Reprosil Pur C18 AQ beads (3 μm, 120 Å particles) at a flow rate of 250 nL/min with a 90-min linear gradient of 2–40% of Buffer B (98% ACN, 0.1% FA). The mass spectrometer was operated with a top 20 DDA data using positive ion mode to acquire the full profile of MS scans. The acquisition mode consisted of a 250 ms survey MS scan of 400–1500 *m/z*, followed by an MS/MS scan of 100–1500 *m/z* for a 100 ms acquisition time. The fragmented precursors were then added to a dynamic exclusion list for 20 s, excluding any singly charged ions from the MS/MS analysis.

#### 4.2.3. Data Independent Acquisition (DIA)—SWATH-MS/MS

The same MS instrument using the identical LC-MS/MS setup, as above, was operated for the targeted data extraction of SWATH-MS/MS acquisitions from iRT spiked peptide solutions. In SWATH-MS/MS mode, the instrument was tuned for the selection of 32 overlapping windows with 25 *m/z* effective isolation width, covering the mass range of 350–1250 *m/z*, with a dwell time of 250 ms in high-sensitivity mode. SWATH-MS/MS scans were performed with a dwell time of 100 ms to cover the mass range of 100–1500 *m/z* with a cycle time of 3.3 s.

### 4.3. LC-MS/MS Data Analysis

#### 4.3.1. Spectral Library Generation

DDA MS data generated from LNCaP treatment/control and C4-2B protein samples were searched by ProteinPilot^TM^ software (Version 5.0.1, AB SCIEX) using the Paragon™ Algorithm against the full non-redundant, canonical human genome as annotated by UniProtKB/Swiss-Prot. Search parameters were as follows: Sample type: Identification; Cys Alkylation: Iodoacetamide; Digestion: Trypsin; Instrument: TripleTOF 5600+; Species: None; Search effort: Thorough ID; Results Quality: Detected protein threshold [Unused ProtScore (Conf)]  ≥  0.05 with FDR.

#### 4.3.2. SWATH-MS/MS Data Analysis

SWATH-MS/MS raw data was processed against the above created spectral library. Extracting peak areas and scoring was performed using the PeakView^®^ SWATH micro app (Version 2.1, AB SCIEX). Shared peptides were excluded while importing the spectral library. Minimum number of 3 transitions (fragment ions) and 3 precursor ions with >90% confidence were used for SWATH-MS/MS data processing. Peak areas were extracted for peptides with <1% FDR. Peak area of fragment ions was used for peptide quantification, whereas the mean value of the peptide quantities was used to quantify proteins.

### 4.4. RNA Isolation

Total RNA was extracted from PCa cells using RNAeasy Mini Kit according to the standard protocol (Qiagen, Hilden, Germany). RNA concentration and purity were measured using NanoDrop^TM^1000 (Thermo Scientific).

### 4.5. Real-Time Quantitative PCR (qRT-PCR) Analysis

RNA (1 μg) was reverse transcribed to cDNA using SuperScript III Reverse Transcriptase (Invitrogen, Carlsbad, CA, USA). Quantitative RT-PCR was performed in MicroAmp^®^ Optical 384-Well Reaction Plate with barcode (Applied Biosystems, Foster City, CA, USA) using the ViiA7 Real-Time PCR system (Applied Biosystems). Each reaction contained 1X final concentration of SYBR Green PCR Master Mix 2X (Life Technologies), 50 nM forward and reverse primer, 2.0 μL of diluted cDNA (1:5) and nuclease-free water to a final volume of 8 μL. The cycling parameters were 95 °C for 10 min, 40 cycles of 95 °C for 15 s and 60 °C for 1 min followed by a dissociation step. Relative fold expression was performed by the comparative CT (ΔΔCT) method.

### 4.6. RNAseq Analysis

Total RNA extracted from the androgen and anti-androgen treated LNCaP cell lines used for the RNAseq analysis performed through the Australian Genome Research Facility (AGRF). Three independent RNAs for each treatment were pooled and used for the ribosomal depletion. Paired-end sequencing (100 bp resolution) has performed on the Illumina HiSeq platform using the Illumina TruSeq strand-specific protocol (Life Technologies). RNAseq was carried out at 100 bp resolution, paired end, and strand specific. Raw data was analysed by mapping RNAseq reads using Tophat2 (hg19 assembly) [70], and logFC and *p* values for the gene expression analysis were determined using the edgeR program [71].

### 4.7. Statistical Analysis

Exported SWATH-MS/MS peak area values were statistically analysed using the MetaboAnalyst 4.0 online tool [72]. First, peak area values of each protein, obtained from the three replicates were averaged, normalized (based on the sum) and log_2_ transformed. PCA analysis was performed for the first three principal components using log_2_ transformed normalized areas. For the Heatmap visualization, normalized SWATH areas were clustered based on Ward algorithm and row/column orders were set by applying a hierarchical clustering based on Euclidean distance. Fold change (FC) expression between each treatment group and the log_2_ transformation of FC (Log_2_FC) was calculated. A threshold of |log_2_FC| ≥ 1 was used to identify DEPs, whereas |logFC| ≥ 1.5 threshold was used identify DEGs. *p* < 0.05 was considered significant.

### 4.8. Network Analysis

SWATH-MS/MS Proteomic datasets and RNAseq datasets were analysed using Ingenuity Pathway Analysis (IPA) (Qiagen, http://www.ingenuity.com) and list of molecular networks were generated based on comparing imported datasets and Ingenuity^®^ Knowledge Base. A score (*p*-score = −log10 (*p*-value)) according to the fit of the set of supplied protein/genes and molecular networks stored in the Ingenuity Knowledge Base were generated. Both direct and indirect relationships were considered and only DEPs/DEGs with a *p* < 0.05 were considered for the analysis.

### 4.9. Upstream Regulatory Analysis

IPA upstream regulator analysis was used to predict the activation/inhibition of upstream transcriptional regulators from the datasets based on the literature and compiled in the Ingenuity^®^ Knowledge Base. A *p*-value was computed based on significant overlap between known targets regulated by the transcriptional regulator and the molecules in the dataset. The activation Z-score algorithm was used, to make predictions on the upstream regulator. The expression of the upstream regulator itself and only DEPs/DEGs with a *p* < 0.05 were considered for the analysis.

### 4.10. Gene Set Enrichment Analysis

GSEA was used to determine over-represented GO-BPs enriched by DEPs/DEGs identified in androgen/anti-androgen treated LNCaP cell models [10]. GSEA converted the submitted proteins/genes into Entrez genes on the MSigDB v7.1 [73,74,75]. The analysis evaluated the overlap of provided DEPs/DEGs with human MSigDB GO-BP gene sets and estimated the statistical significance at FDR/*q* < 0.05.

### 4.11. KEGG Mapper Analysis

KEGG mapper tool was used to map DEPs/DEGs in to KEGG pathways that are dysregulated in AD LNCaP cell model in response to anti-androgen treatment and AI C4-2B cell model. The tool converted the submitted proteins/genes into KEGG GENES ID and matched with the human specific KEGG pathway maps enabling interpretation of cellular functions and other high-level features [76,77]. Pathway lists and categories were downloaded from the KEGG database and different colour codes were used to demonstrate treatment specific and common DEPs/DEGs identified [78].

### 4.12. Validating the Expression of Differentially Regulated Genes/Protein

Differentially expressed protein/genes identified in PCa cell models were revaluated by measuring their expression in Grasso PCa clinical datasets [79] using the online Oncomine^TM^ database (https://www.oncomine.org) [80]. Protein expression in PCa clinical/tissue specimens were analyzed by the PCa immunohistochemistry (IHC) datasets using the Human Protein Atlas (HPA) database (http://www.proteinatlas.org) [81,82].

## 5. Conclusions

The current study tested the hypothesis that molecules dysregulated in response to ATT enable the overrepresentation of adaptive molecular pathways supporting PCa treatment escape. Using SWATH-MS strategy, we highlighted anti-androgen induced key proteomic signatures that are involved in transcription/translation regulation, energy metabolism, cell growth and death, cell communication and signaling cascades that might potentially evolve to treatment resistance and androgen-independence in PCa cells. Following further validation, these proteomic signatures and associated pathways can be co-targeted to improve the efficacy of existing treatments and develop novel treatment strategies to attenuate disease progression to CRPC.

## Figures and Tables

**Figure 1 cancers-13-00715-f001:**
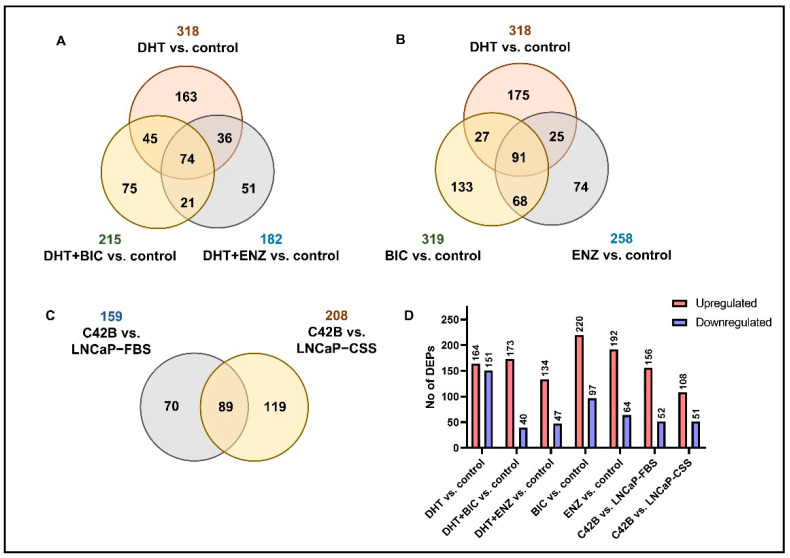
Summary of the SWATH-MS proteomic analysis. (**A**) The number of DEPs overlapped between DHT, DHT + BIC and DHT + ENZ treatments compared to control treatment. (**B**) The number of DEPs identified in DHT, BIC and ENZ treatments compared to control treatment. (**C**) The number of DEPs overlapped between C4-2B vs. LNCaP-FBS and C4-2B vs. LNCaP-CSS. (**D**) The bar chart represents the number of upregulated and downregulated DEPs in all treatment comparisons.

**Figure 2 cancers-13-00715-f002:**
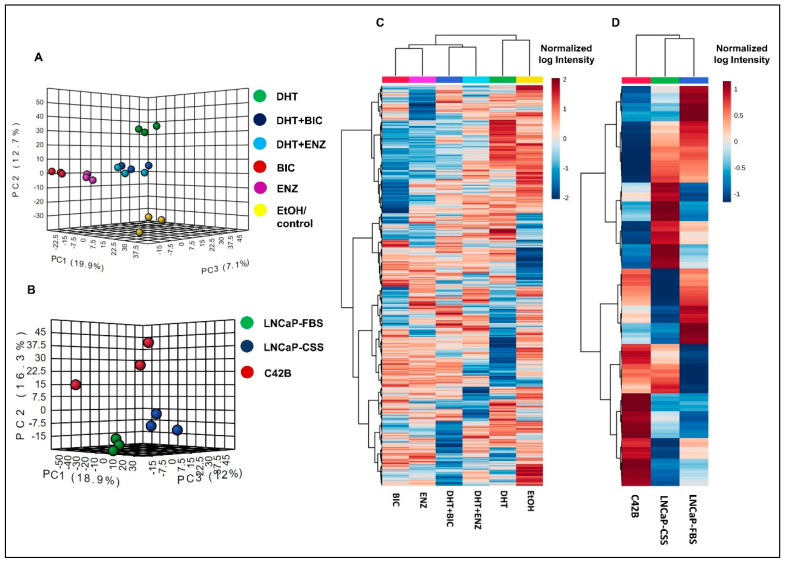
Proteomic profiles in LNCaP cells with response to androgen/anti-androgen treatment. The graph demonstrates the 3D principal component scores plot for the proteomic profiles of (**A**) androgen and anti-androgen treated LNCaP cells and (**B**) C4-2B and LNCaP cells. Hierarchical clustering analysis of (**C**) androgen and anti-androgen treated LNCaP cells and (**D**) C4-2B and LNCaP cells. For the Heatmap visualization, normalized SWATH areas were clustered based on Ward algorithm, and row/column orders were set by applying a hierarchical clustering based on Euclidean distance.

**Figure 3 cancers-13-00715-f003:**
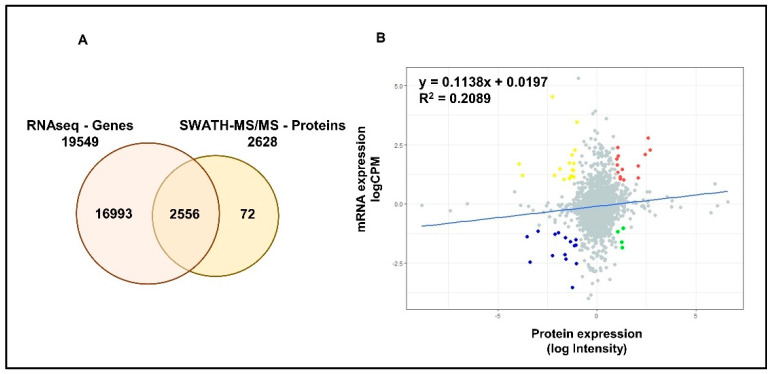
Correlation between mRNA and protein expression in LNCaP cells. (**A**) Venn diagrams demonstrate the number of genes identified by RNAseq and SWATH-MS analysis of LNCaP cells. A total of 2556 common elements were identified by both experiments. (**B**) Scatter plot represent the correlation between protein expression measured as log Intensity and respective mRNA expression measured as log counts per million (CPM) in LNCaP cells. R^2^ value indicates the variability between the mRNA and protein expressions. Slope indicates the linear regression coefficient. Red: Genes showing high mRNA and high protein expressions; Blue: Genes showing low mRNA and low protein expressions; Yellow: Genes showing high mRNA and low protein expression; Green: Genes showing low mRNA and high protein expression.

**Figure 4 cancers-13-00715-f004:**
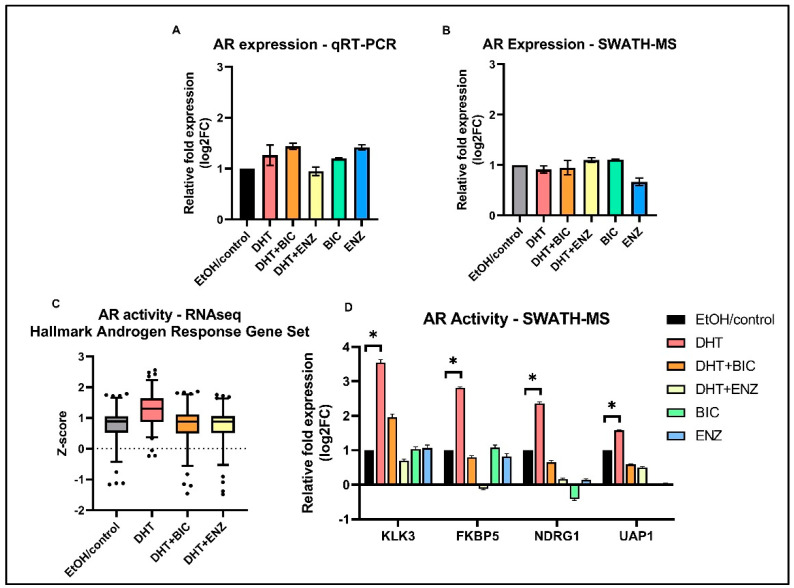
AR expression and AR transcriptional activity in androgen and anti-androgen treated LNCaP cells. Relative fold AR expression measured in (**A**) transcript level by qRT-PCR assay and (**B**) protein level by SWATH-MS analysis. (**C**) Standardized gene expression (z-score) of the MSigDB Hallmark “Androgen Response” gene set in response to DHT, DHT + BIC, DHT + ENZ and EtOH/control treatments. (**D**) Relative fold protein expression of four ARGs; KLK3, FKBP5, NDRG1, UAP1 in response to DHT, DHT + BIC, DHT + ENZ, BIC, ENZ and control treatments. All measurements are normalized to control treatment. Error bars represent the means ± SD, * *p* < 0.05.

**Figure 5 cancers-13-00715-f005:**
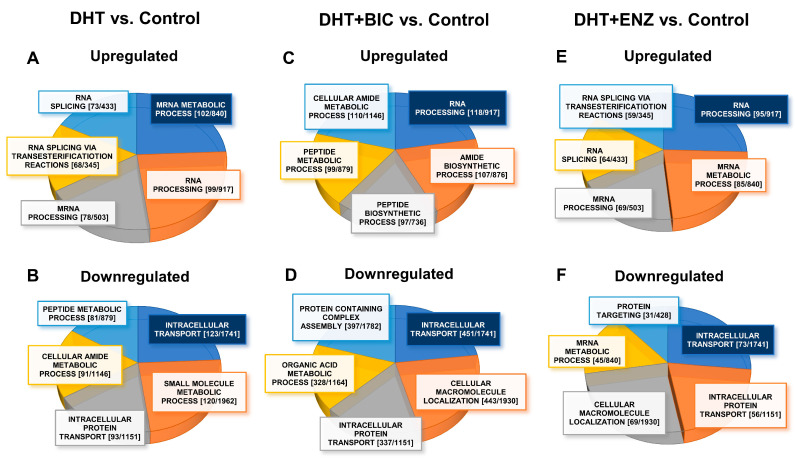
GO-biological processes overrepresented by the proteins up-/down-regulated in response to androgen and anti-androgen treatment. The pie charts represent the GO-biological processes significantly dysregulated (FDR/q < 0.05) in DHT treatment: (**A**) upregulated (**B**) downregulated; DHT + BIC treatment: (**C**) upregulated (**D**) downregulated; and DHT + ENZ treatment: (**E**) upregulated (**F**) downregulated, compared to control treatment. Highlighted (Blue) biological process represent the most significantly dysregulated GO-biological process. The size of each section represents the associated number of proteins associated for each GO biological process. Each GO-biological process is given with [the number of DEPs (k)/number of molecules GO-BP gene set (K)].

**Figure 6 cancers-13-00715-f006:**
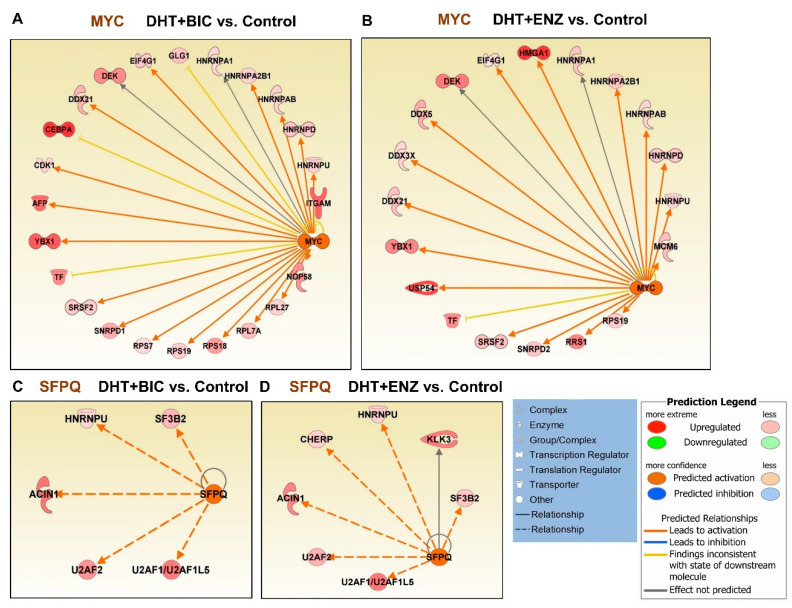
Upstream regulatory analysis for the differentially expressed genes/proteins identified by anti-androgen treatments. Regulatory effect of MYC upstream regulator on target genes in response to (**A**) DHT + BIC treatment, (**B**) DHT + ENZ treatment compared to control treatment. Regulatory effect of SFPQ upstream regulator on target genes in response to (**C**) DHT + BIC treatment, (**D**) DHT + ENZ treatment compared to control treatment. Colour in the target genes represent their upregulation or downregulation while colour in the upstream regulator represent their predicted activation or inactivation. Colour intensity represents the relative magnitude of change in expression. Edges are coloured to indicate the expected direction of regulation.

**Figure 7 cancers-13-00715-f007:**
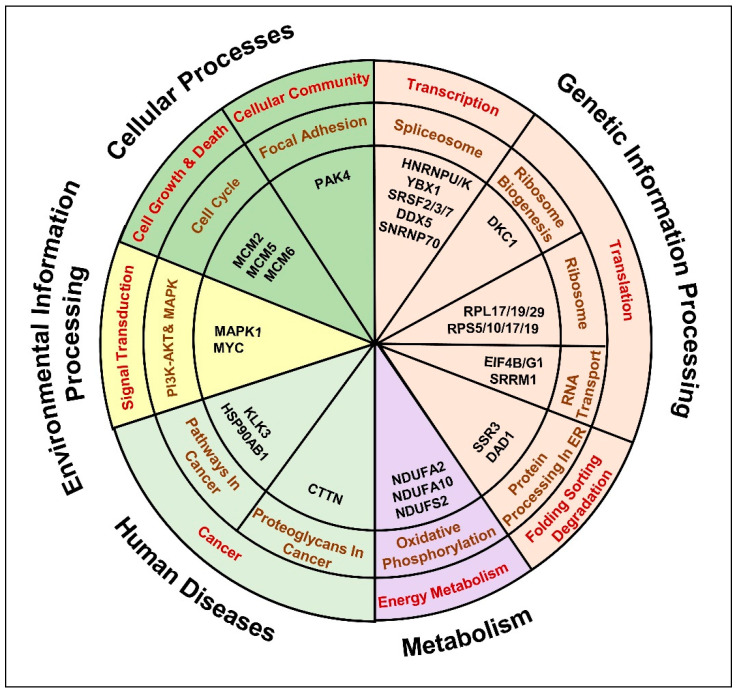
Anti-androgen induced molecular pathways that denote the insurgence of metastatic, androgen independent PCa phenotype. The chart represents the key dysregulated proteins common to anti-androgen administered androgen-sensitive LNCaP cells and C4-2B androgen-independent cells identified by the comparative bioinformatics analysis of proteomic and transcriptomic data. Molecules were distributed in to 11 KEGG pathways possibly affected in cancer.

**Figure 8 cancers-13-00715-f008:**
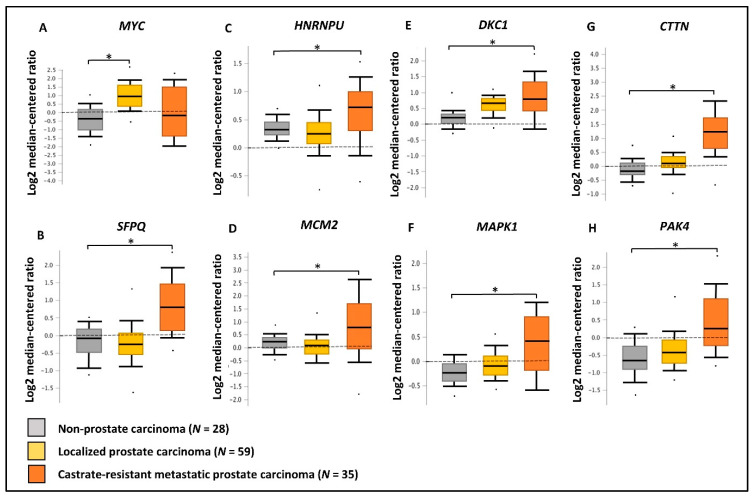
Validation of gene expression using PCa clinical transcriptomic datasets. Comparison of the expression of (**A**) *MYC*, (**B**) *SFPQ*, (**C**) *HNRNPU*, (**D**) *MCM2*, (**E**) *DKC1*, (**F**) *MAPK1*, (**G**) *CTTN* and (**H**) *PAK4* genes between non-prostate carcinoma (*n* = 28), localized prostate carcinoma (*n* = 59), and castration-resistant prostate carcinoma (*n* = 35) tissue microarray data using Grasso PCa clinical dataset available in Oncomine^TM^ database. Log2 median cantered ratio; * *p* < 0.05.

**Figure 9 cancers-13-00715-f009:**
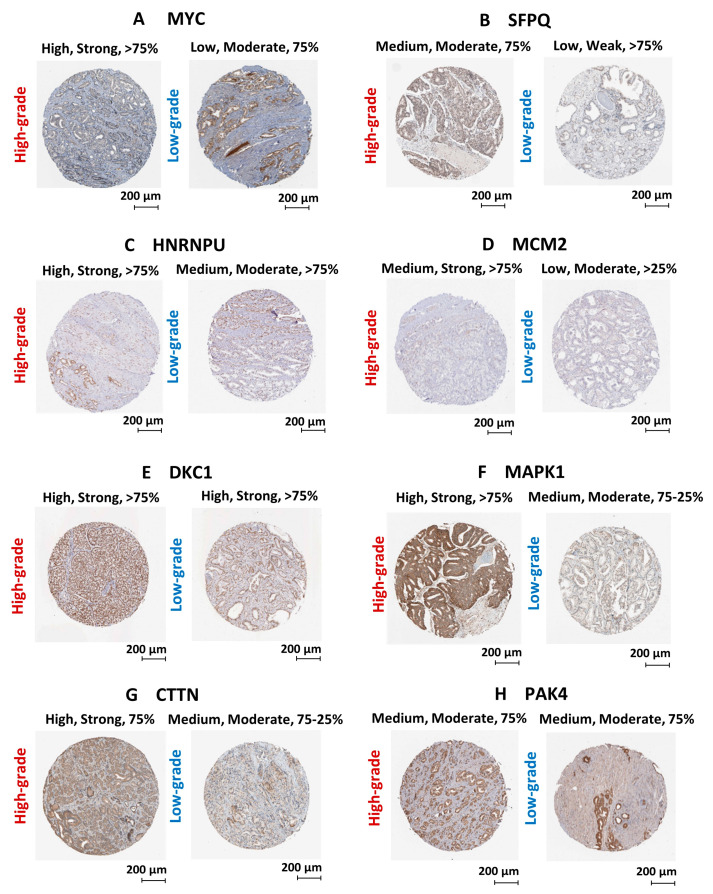
Validation of protein expression using PCa clinical tissue immunohistochemical datasets. Immunohistochemical images of PCa clinical tissues specimens retrieved from the HPA database representing the protein expression of (**A**) MYC, (**B**) SFPQ, (**C**) HNRNPU, (**D**) MCM2, (**E**) DKC1, (**F**) MAPK1, (**G**) CTTN and (**H**) PAK4 protein targets. Each protein is demonstrated with two staining images representing high-grade and low-grade tissue specimens. Parameters that used to annotate each staining include level of antibody staining (High/Moderate/Low), level of staining intensity (Strong/Moderate/Weak) and fraction of immunoactive cells (>75%/75–25%/>25%).

## Data Availability

The data presented in this study are available on request from the corresponding author.

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
