# Peer review of "SWATH-MS Based Proteomic Profiling of Prostate Cancer Cells Reveals Adaptive Molecular Mechanisms in Response to Anti-Androgen Therapy"

_cancers, 2021, doi:10.3390/cancers13040715_

Round 1

Reviewer 1 Report

1) The study confirms on a molecular basis the superiority of Enzalutamide over Bicalutamide. Even if it was yet demonstrated on a clinical basis by trials as STRIVE and TERRAIN in the metastatic CRPC setting (data not mentioned in the paper);

2) As reported in your introduction (lines 52-53) "ADT has been progressively supplemented with inhibitors of of androgen synthesis and  AR antagonists/antiandrogens, therefore termed as androgen targeted therapy (ATT)". Since the title of the publication also refers to ATT (not only antiandrogen therapy), is there a reason why the study of the molecular alterations induced by Abiraterone Acetate (an inhibitor of androgen synthesis) has not been performed? This might give some suggestions on differences in efficacy between the two targeting mechanisms of the AR pathway, also in relation to possible sequencing strategies.

3) There are preclinical and clinical studies demonstrating the synergy between PARP-inhibitors (such as Olaparib) and drugs inhibiting androgen synthesis or AR antagonists (for example, see: Asim M, et al. Nat Commun 2017; Schiewer MJ, et al. Cancer Discov 2012; Clarke N, et al. Lancet Oncol 2018). Specifically: the inhibition fo the transcriptional activity of AR leads to a phenotype comparable to a BRCAness one, susceptible of treatment with PARP-i. Naturally, the presence of BRCA1/2 or other HRR genes mutations confers greater efficacy to PARP-inihibitors (for example, see: De Bono J, et al. NEJM 2020). Data about the expression of HRR genes during exposure to ATT drugs are missing in this paper.

Reviewer 2 Report

The authors present an gene and protein expression analysis of LNCaP cells treated with Bicalutamid and Enzalutamid +/- DHT as well as in LNCaP C4/2 subline generated under castration conditions. They found differences in both gene and protein expression profiles depending on the treatment and/or disease stage (hormone-sensitive vs. castration-resistent). They validate their findings using publically available databases.

In total, the authors should be congratulated for their efforts contributing to this important topic. I have a few comments:

1. p.10, row 277: do you mean "high" instead of higher?

2. p. 12, row 333: a verb must be added to this sentence

3. In the section Discussion, a para describing limitations of the study must be added

4. Why did you use bicalutamid (partial agonistic effects, not very selective, 1st generation AR blocker) in your study? In face of recent advances in therapeutic paradigm of metastatic hormone-sensitive PCa, this drug is almost not being used any more. Why did not you take on apalutamide or darolutamide as second-generation androgen receptor blocker? Or (even more clinically relevant in order to select the most promising second-line treatment) docetaxel? 

5. It would be interesting to see the effects of another type of ATT on transcriptom and proteom by using abiraterone acetate.

6. Generalization of these findings using only one baseline cell type is limited. Did you compare you data to another cell line containing AR like e.g. VCaP?  

Round 2

Reviewer 1 Report

I appreciated the additions to the work made by the authors, whom I thank for the work done and for the answers given to the comments made. Currently, the work is more complete on the topics raised in the first review.

Reviewer 2 Report

The authors responded to my questions sufficiently.